# Characterization of the *RAD52* Gene in the Budding Yeast *Naumovozyma castellii*

**DOI:** 10.3390/genes14101908

**Published:** 2023-10-05

**Authors:** Humberto Itriago, Zubaida Marufee Islam, Marita Cohn

**Affiliations:** Department of Biology, Genetics Group, Lund University, Sölvegatan 35, SE-223 62 Lund, Sweden

**Keywords:** *RAD52*, DNA damage, DNA repair, budding yeast, *Naumovozyma castellii*, *rad52Δ* null mutant, UV irradiation, hydroxyurea, bleomycin

## Abstract

Several sources of DNA damage compromise the integrity and stability of the genome of every organism. Specifically, DNA double-strand breaks (DSBs) can have lethal consequences for the cell. To repair this type of DNA damage, the cells employ homology-directed repair pathways or non-homologous end joining. Homology-directed repair requires the activity of the *RAD52* epistasis group of genes. Rad52 is the main recombination protein in the budding yeast *Saccharomyces cerevisiae*, and *rad52Δ* mutants have been characterized to show severe defects in DSB repair and other recombination events. Here, we identified the *RAD52* gene in the budding yeast *Naumovozyma castellii*. Our analysis showed that the primary amino acid sequence of *N. castellii* Rad52 shared 70% similarity with *S. cerevisiae* Rad52. To characterize the gene function, we developed *rad52Δ* mutant strains by targeted gene replacement transformation. We found that *N. castellii rad52Δ* mutants showed lowered growth capacity, a moderately altered cell morphology and increased sensitivity to genotoxic agents. The decreased viability of the *N. castellii rad52Δ* mutants in the presence of genotoxic agents indicates that the role of the Rad52 protein in the repair of DNA damage is conserved in this species.

## 1. Introduction 

DNA double-strand breaks (DSBs) are lethal threats to every organism because they compromise genome stability. Some DSBs are intermediates of programmed recombination events, while others can occur spontaneously during the metabolic reactions of the cell or by external damaging agents [1]. There are two main pathways for the repair of this type of DNA damage: non-homologous end joining (NHEJ) and homologous recombination. The latter is considered a high-fidelity repair mechanism since it involves the use of a homologous duplex template to repair the damage inflicted at the site; thus, these mechanisms preferentially take place during the S and G2 phases of the cell cycle [2,3,4]. The homologous recombination pathway is mediated by a series of proteins belonging to the *RAD52* epistasis group of genes, which includes *RAD50*, *RAD51*, *RAD52*, *RAD54*, *RAD55*, *RAD57*, *RAD59*, *RDH54*, *MRE11* and *XRS2* [5]. The members of this group have primarily been identified in the budding yeast *Saccharomyces cerevisiae* and characterized as genes that make the cell extremely vulnerable to agents that generate DSBs, especially ionizing radiation [6].

*RAD52* is the defining member of the group for its role in different types of homologous recombination events [4]. For DNA repair by classic double-strand break repair (DSBR), nucleolytic degradation of the 5′ strand generates a 3′ single-stranded DNA end that is quickly bound by the ssDNA binding protein Replication Protein A (RPA) (Figure 1) [7]. As a recombination mediator, Rad52 has the ability to displace RPA and recruit Rad51 to the site of the DNA damage. Multiple proteins are associated with and cooperatively bind the ssDNA. The DNA-Rad51 filament is capable of promoting complementary strand annealing to the donor duplex. Strand invasion generates a displacement (D) loop structure and invading 3′ ssDNA can serve as a primer for DNA synthesis. While new bases are added to the invading 3′ end, Rad52 captures the second end of the break and anneals it to the D-loop. After extension by DNA polymerase, the ends are ligated, generating a double Holliday junction that, when resolved, leads to the repair of the DSB [4,7].

Alternatively to DSBR, after D-loop formation and the extension of the invading DNA, the ssDNA can be dissociated from the D-loop and complementarily annealed to the second end at the site of the break, mediated by the strand annealing function of Rad52 [4,7]. In this mechanism, termed synthesis-dependent strand annealing (SDSA), gap filling by DNA polymerase and ligation concludes the DNA repair process. The ssDNA annealing activity of Rad52 can also mediate non-canonical homology-directed repair mechanisms such as single-strand annealing (SSA) [7]. Moreover, the mechanisms of telomere maintenance in the absence of telomerase have been shown to be dependent on Rad52-mediated break-induced recombination and can occur in a Rad51-dependent or Rad51-independent mechanism, the latter requiring the function of Rad59 and Rad50 [8,9]. 

The Rad52 protein’s N-terminal structure is highly conserved from yeast to humans [10,11,12]. The yeast Rad52 protein contains three functional domains necessary for its mediator and DNA annealing functions [4]. The N-terminal domain of the protein is the most evolutionarily conserved region and it contains DNA-binding domains, sites of interaction with the paralog Rad59 and two self-association domains that allow for the association of seven or more subunits into a characteristic ring structure [4,13]. The middle region is necessary for the nuclear transport of the protein and the interactions with the RPA components [14]. The C-terminal domain can bind DNA but most importantly allows for the association with Rad51 [15].

The budding yeast *Naumovozyma castellii* is a closely related species to *S. cerevisiae*. The haploid cells exhibit either of the two mating types *MAT*a and *MAT*α and can form hybrids with *S. cerevisiae* and other closely related yeasts containing this conserved mating type system [16]. *N. castellii* is positioned after the whole genome duplication event that took place in the evolutionary lineage of Saccharomyces, an event that was followed by the large-scale loss of duplicated genes and chromosomal rearrangements, resulting in the *N. castellii* genome size of 11.2 Mb distributed on 10 chromosomes in haploid cells [17]. Notably, *N. castellii* has contributed extensively to comparative genomic studies and the discovery of its unique centromere DNA elements unveiled remarkably rapid evolutionary changes in centromeres [18]. The susceptibility to genetic manipulation and the development of stable heterothallic haploid strains has promoted *N. castellii* as a model system for functional molecular biology studies, and it has been extensively used as a model organism in telomere biology research due to its beneficial telomere characteristics. The homogeneous telomeric repeats are synthesized by highly processive telomerase activity, where the enzyme performs translocation on the telomerase RNA template to produce several repeats per annealing cycle [19,20].

Rad52 is a key player in multiple DNA double-strand break repair pathways, which are important core mechanisms that connect to other pathways to maintain genomic stability, thus making *RAD52* one of the most interesting genes to study and to delineate its function in this newly established model species. Here, we aimed to identify and characterize the *N. castellii RAD52* homolog. To this end, we first performed a bioinformatic analysis of the primary amino acid sequence in comparison to *S. cerevisiae* Rad52. We found that the two proteins share a high degree of identity (60.6% identity, 70.3% similarity) and, most importantly, the conservation of essential residues at the three functional domains described above. To characterize the *RAD52* gene function, we developed *rad52Δ* null mutants by targeted gene replacement. We report that haploid *rad52Δ* mutants have a slightly lower growth rate and somewhat larger cell size compared to the WT parental strain. The *rad52Δ* strains are mildly sensitive to UV irradiation but are highly sensitive to two genotoxic agents known to indirectly or directly generate DSBs: hydroxyurea and bleomycin, respectively. These results indicate that *N. castellii rad52Δ* mutants have an impaired capacity to repair DSBs, as has been described for *S. cerevisiae rad52Δ* mutants.

## 2. Materials and Methods

### 2.1. Strains and Cell Culture

*N. castellii* was previously called *Saccharomyces castellii* or *Naumovia castellii*. The *N. castellii* strains used in this study were the parental wild-type strain YMC48 and the derived YMC490-494 strains (Table 1). Strains were grown at 25 °C in YPD medium containing 1% (*w*/*v*) yeast extract, 2% (*w*/*v*) peptone and 2% (*w*/*v*) glucose or plates additionally containing 2% (*w*/*v*) agar, unless stated otherwise. Ura-dropout plates contained 26.7 g/L minimal SD base, 0.77 g/L ura-DO supplement and 2% agar.

Cell passaging was performed by streaking a single colony onto fresh YPD plates and, after 48 h incubation at 25 °C, single colonies were re-streaked onto new YPD plates. Re-streaking was repeated to obtain up to 16 subsequent passages. Each passage corresponded to 20–25 generations [16,21].

Total cell counts were performed with the Nucleocounter^®^ YC-100™ system (Chemometec, Allerød, Denmark) following the manufacturer’s instructions. For growth assays, 50 mL liquid YPD medium cultures were started at a cell density of 3 × 10^6^ cells/mL from an overnight culture inoculated with a single colony. The cultures were grown at 25 °C, 200 rpm shaking, for 8 h. Measurements of cell density were taken directly after inoculation and then at two-hour intervals. 

### 2.2. Transformation 

The *RAD52* deletion cassette containing the klURA3 cassette flanked by the upstream and downstream sequences of the *N. castellii RAD52* locus was designed by the authors and manufactured by GenScript. The klURA3 cassette contains the *Kluyveromyces lactis URA3* gene flanked by 143 nt long direct repeats, allowing the subsequent pop-out of the *URA3* gene [16,21]. The *RAD52* deletion construct was PCR-amplified with the forward primer (52KO-F) 5′-TGGTCGTCTTGTCCAATGCT-3′ and reverse primer (52KO-R) 5′-GTTGGGAATGGATTGGACCCT-3′, using Phusion Polymerase (Thermo Fisher Scientific, Waltham, MA, USA), according to the manufacturer’s instructions, in 25 cycles: 30 s annealing time at 61 °C and 1 min elongation time. The amplification products were ethanol-precipitated and resuspended in 10 mM Tris-HCl pH 8 and quantified using a NanoDrop spectrophotometer.

YMC48 cells were transformed with the deletion construct using the lithium acetate transformation method, as previously described [16,22]. Briefly, a 50 mL YPD culture was inoculated at 3 × 10^6^ cells/mL from an overnight culture grown from a single colony. The culture was grown at 25 °C and 200 rpm shaking, until a cell density of 1 × 10^7^ cells/mL was reached. The cells were harvested by centrifugation at 2500× *g* for 5 min, washed with sterile water and resuspended in 25 mL transformation buffer I (0.1 M lithium acetate, 10 mM Tris–HCI, 1 mM EDTA pH 7.5). Cells were pelleted and resuspended in 250 μL transformation buffer I. One µg of the DNA construct was mixed with 20 µg freshly denatured salmon sperm DNA in a total volume of 10 μL and mixed with 50 μL competent cells and 300 μL transformation buffer II (40% PEG 3350 in transformation buffer I). The transformation mix was incubated at 25 °C for 30 min and then heat-shocked at 42 °C for 10 min. After the heat shock, the cells were washed twice with sterile water and resuspended in 1 mL YPD. After an hour of incubation at 25 °C, the cells were spread on Ura-dropout plates and incubated at 25 °C for 2–3 days.

### 2.3. Genotype Analysis of rad52Δ Transformants

For the amplification of the *RAD52* locus, the flanking primers R52Fl-F (5′-AGGCCAAAACAATCCCCCAA-3′) and R52Fl-R (5′-TCTCTAGGAATGCCGCCAAC-3′) were used in the Phusion Polymerase reaction following the manufacturer’s instructions: 30 s annealing time at 66 °C and 1.15 min elongation time. To confirm the correct insertion of the klURA3 cassette, amplification with the R52Fl-F primer and the klURA3 cassette-specific primer klURA3-R (5′-TTGTGAAAGCCAGTACGCC-3′) was performed: 30 s of annealing at 59 °C and 1.15 min elongation time. Additionally, a separate reaction was performed with the klURA3 cassette primer klURA3-F (5′-GTTGAAGTGAGTGTTGCACC-3′) and R52Fl-R: 30 s annealing at 60 °C and 1.15 min elongation time. PCR products were resolved in 0.6% agarose gels in 0.5× TBE buffer (45 mM Tris-borate, 1 mM EDTA).

### 2.4. Spot Assays

Whole colonies from the parental WT and *rad52Δ* strains were grown overnight in 5 mL YPD, at 25 °C with 200 rpm shaking. Serial 10-fold dilutions were prepared after measuring the cell density of the cultures as described above and spotted onto YPD plates and incubated for 2–3 days at 25°, unless stated otherwise. UV irradiation was performed with a Spectrolinker™ XL-1000 (254 nm bulbs) (Spectronics Corporation, New York, NY, USA). Hydroxyurea (H8627, Sigma Aldrich, St. Louis, MO, USA) and bleomycin (BP9971, Sigma Aldrich, St. Louis, MO, USA) were prepared in sterile milliQ H_2_O. Hydroxyurea and bleomycin plates were created by adding the appropriate amounts of the respective chemicals into YPD plate media before pouring the plates. Pictures of the plates were taken in a GelDoc XR+ (Bio-Rad, Hercules, CA, USA).

### 2.5. Survival Curves

For cell viability assessments, cells from a single colony were grown in 25 mL YPD at 25 °C with 200 rpm shaking until the culture reached the logarithmic growth phase. Total cell counts were performed using a Brand™ Bürker counting chamber (Fisher Scientific, Waltham, MA, USA). Cells were diluted to a final concentration of 2 × 10^3^ cells/mL using serial 10-fold dilutions in 1× TNE (1 mM Tris-HCl, 10 mM NaCl, 0.1 mM EDTA, pH 7.4). From the suspension, 200 cells were plated (100 µL) on solid YPD medium and plates were incubated at 20, 25 and 30 °C, for 3 days. Colony forming units were counted and photographed in a GelDoc XR+ (Bio-Rad, Hercules, USA).

Quantitative survival curves were obtained by plating 200 cells on solid YPD medium containing hydroxyurea or bleomycin at different concentrations, or YPD plates that were irradiated with different doses of UV light. For each treatment, 1000 cells were plated in total (5 technical replicates) and cells from 3 individual colonies per strain were tested as biological replicates. Colony forming units (CFU) were counted after 3 days of incubation at 25 °C and photographed in a GelDoc XR+. For plates containing hydroxyurea, the count was performed on the color-inverted photograph with the contrast adjusted to eliminate any background of microcolonies. The number of CFU was counted for each treatment, normalized to the non-treated controls and plotted in diagrams as the percentage of survivors at the respective dose.

### 2.6. Colony and Cell Imaging

Cells were grown on YPD plates for 2 days at 25 °C and the colonies were photographed with an Olympus SZ61 stereo microscope equipped with a CAM-E3CMOS6.3 camera using the ImageView software, and the size measurement performed was using the 3-point circle tool. For the collection of images of the cells, phase contrast microscopy was performed using a Zeiss Observer Z1 inverted microscope Ph3 stage with the Pln Apo 100X/1.4 objective by immobilizing cells from an overnight culture in a 1 mm layer of 1% agarose in PBS. Cells were photographed using the Hamamatsu ORCA-Flash4.0 LT + C11440 camera and the ZEN software. For cell measurement, the cells were photographed with an Olympus CX43 microscope equipped with a CAM-E3CMOS6.3 camera using the ImageView software, and the size measurement was performed with the 5-point ellipsis tool. All figures were prepared with Adobe Illustrator. 

### 2.7. Bioinformatic Analysis

The sequence of the *N. castellii RAD52* ORF was retrieved from the KEGG genome database (NCAS_0H02830) [23]. The sequences of *S. cerevisiae* Rad52 (NCBI-ProteinID P06778) and *N.castellii* Rad52 (NCBI-ProteinID XP_003677940.1) were aligned by ClustalW. To retrieve sequences similar to *N. castellii* Rad52, we performed homology probing with BLAST v2.12.0 using the *N. castellii* sequence as a query against the SWISS-PROT database (accessed October 2021), with default parameters (e-value of <10^−6^). All subject sequences were retrieved using the ENTREZ package implemented in Biopython v1.79 [24]. Sequences were aligned using MUSCLE v5.1 and ambiguously aligned regions were trimmed using trimal v1.4.1 (-gappyout). SnapGene was used to visualize the multiple sequence alignment. 

To study the sequence of *N. castellii* Rad52 against closely related homologs, we retrieved the primary sequences from eggNOG v5.0 ENOG503S0VW corresponding to the Saccharomycetaceae family. Since many of these proteins have not been characterized, we utilized HMMR v3.3.2 (hmmsearch) to detect the regions of the protein that matched the hidden Markov model corresponding to the “DNA repair and recombination protein Rad52, Rad59” family described in the PANTHER database (PTHR12132). The defined regions of these sequences were aligned using MUSCLE and trimmed and visualized using SnapGene as above. Python scripts are available on request. 

## 3. Results and Discussion

### 3.1. DNA Sequence Analysis of the N. castellii RAD52 Gene

*RAD52* gene function is essential for multiple mechanisms of homologous recombination, including homology-directed DNA repair. The sequence of a 1356 nucleotide open reading frame (ORF) hypothesized to contain the *RAD52* gene (NCBI-GeneID 11528895) was annotated on the genome database of the budding yeast *N. castellii* based on the amino acid similarities to the family of DNA repair and recombination motifs derived from the conserved structure of Rad52/Rad22 proteins [25]. To identify and characterize the protein encoded in this region, we performed local protein homology probing (BLAST) with the protein sequence as a query against the SWISS-PROT protein sequence database. Indeed, the structure of the hypothetical protein was matched to Rad52 proteins and Rad52 homologs described for different eukaryotic species, including the closely related budding yeast *S. cerevisiae*. 

The *S. cerevisiae* and *N. castellii RAD52* gene sequences share 61.2% nucleotide sequence identity. The Rad52 protein comparison showed significant overall sequence homology at the amino acid level (70.3% similarity and 60.6% identity) (Figure 2). Specifically, we found three highly conserved regions corresponding to regions of the *S. cerevisiae* Rad52 protein described to have different molecular functions. Interestingly, after the multiple sequence alignment of the Rad52 protein sequences of different homologs within the Saccharomycetaceae family, we found these three high-homology regions to be present in all species analyzed (Appendix A). However, the comparison with orthologs of more distant species found by BLAST showed the conservation only of the N-terminal domain, as expected (Appendix A).

At the N-terminal domain, 169 amino acid residues corresponding to *S. cerevisiae* Rad52 residues 52-220 shared 95.9% similarity (91.1% identity) between the two proteins (Figure 2). The N-terminal domain of *S. cerevisiae* has been described to contain a DNA-binding domain and residues that are important for the self-association required for the formation of its characteristic heptameric ring structure [12,14,27]. In this regard, it is worth noting that all amino acid residues previously described to cause loss of function in *S. cerevisiae*—for example, P64L, R85W, A90V, N91A, F94A, R136B and G140C—are conserved in *N. castellii* Rad52, thus providing the conditions for a similar function of the domain [12,28]. Moreover, the N-terminal domain of *S. cerevisiae* Rad52 has also been shown to interact with Rad59, allowing the formation of complexes between these two proteins and RPA or Rad51 [13,29]. The *S. cerevisiae* A89 residue is essential for this interaction and is conserved in the *N. castellii* primary sequence [30]. The functional importance of the N-terminal domain is evident from the fact that the sequence similarity is remarkably high between homologs of phylogenetically widely separated organisms, ranging from yeast to human Rad52 [10,31].

The middle region of the Rad52 protein has been shown to be necessary for the interaction of Rad52 with the RPA protein in *S. cerevisiae* [14]. In this region, 27 amino acid residues corresponding to the residues 290–316 of *S. cerevisiae* Rad52 share 96.3% similarity (92.6% identity) with *N. castellii* Rad52 (Figure 2). Interestingly, this region of homology overlaps with a stretch of acidic amino acids important for the formation of the Rad52 repair center (*S. cerevisiae* residues 288–327). Notably, every residue that, when altered, was shown to decrease the resistance to methyl methanesulfonate (MMS)-induced DNA damage is conserved in *N. castellii* [14]. At the middle of the *S. cerevisiae* protein, the residues 231-PNKRR-235 constitute the pat7 Nuclear Localization Signal (NLS), which consists of a proline (P) followed by 3–4 basic amino acid residues. This NLS was shown to be necessary for the localization of Rad52 in the nucleus, with the core KRR residues being essential for its transport [32] (Figure 2B). *N. castellii* Rad52 lacks a pat7 NLS and instead contains a pat4 NLS sequence, 189-NKRR-192 (four continuous basic residues, Figure 2B). Since the core KRR remains unaltered, it is possible that this pat4 NLS allows for the transport of the protein to the nucleus in *N. castellii*. 

*S. cerevisiae* Rad52 is modified by the ubiquitin-like protein SUMO in three major sites flanking the N-terminal DNA-binding domain of *S. cerevisiae* (lysine residues 43, 44 and 253) [33]. While non-sumoylated Rad52 does not greatly affect the recombination levels of the cells, two variants of Rad52 (mono- and di-sumoylated) become abundant when cells are exposed to genotoxic agents [4,33]. *N. castellii* Rad52 has retained the first major sumoylation site (*S. cerevisiae* lysines 43/44) but is missing the sumoylation consensus site corresponding to *S. cerevisiae* lysine 253. This would imply that *N. castellii* Rad52 would mainly be mono-sumoylated in the presence of genotoxic agents. This could be interesting to study further in this model organism, as Rad52 sumoylation has been shown to modulate recombination-based telomere maintenance in the absence of telomerase activity, favoring Rad51-dependent recombination at the telomeres [34].

At the C-terminal region, 72 amino acids in the 348-419 position of *S. cerevisiae* Rad52 share 90% similarity (69% identity) with *N. castellii* (Figure 2). The C-terminal region of *S. cerevisiae* Rad52 contains the interaction domain for association with the Rad51 protein and has been shown to bind DNA. Two motifs are essential for the interaction with Rad51 in *S. cerevisiae*, 349-FVTA-352 and 409–YEKF-412 [35,36]. In *N. castellii*, these essential residues showed complete amino acid identity. The Rad51 interaction domain has also been mapped to the C-terminal region of the protein in human Rad52 and shown to have similar functions and, additionally, to aid with the self-association of the protein [37,38]. Furthermore, two short stretches of unknown function (*S. cerevisiae* residues 461R-N473 and 486T-N496) are highly conserved in the C-terminal region.

In summary, we identified the sequence of the *N. castellii RAD52* gene and, through a comparison analysis to the well-characterized *S. cerevisiae* Rad52 protein, we were able to determine that the *N. castellii* homologous protein has been highly conserved. Structurally, there is the conservation of essential amino acid residues in the three distinct functional regions that are necessary for the mediator and annealing functions of Rad52, self-association and nuclear transport. 

### 3.2. Deletion of the N. castellii RAD52 by Gene Replacement

To characterize the phenotype of cells lacking the *RAD52* gene, we aimed to develop haploid *rad52Δ* strains by targeted gene replacement (Figure 3A). From our recently developed stable haploid strains, we selected YMC48 (*MATα*, *hoΔ::hphMX4*, *ura3*) as the parental strain for this study [16]. To delete the *RAD52* gene, we designed a DNA construct containing the *Klyuyveromyces lactis URA3* cassette (klURA3) flanked by the upstream and downstream sequences of the *N. castellii RAD52* gene. The klURA3 cassette contains the sequence of the *K. lactis URA3* gene as the marker gene for transformation, flanked by short direct repeats (143 bp) that can later allow the excision of the marker gene by recombination (Appendix A). 

Homologous recombination is the predominant repair mechanism in *S. cerevisiae*, and its high activity allows for the targeted insertion of fragments with homology as short as 30–45 bp for transformation by ends-out recombination. For *N. castellii*, such short sequences are not productive in ends-out targeted gene replacement approaches, while long homology regions (500–1000 bp) give a very efficient result [22]. By the extensive testing of different lengths of homology, we have shown that medium-sized homology regions of 200–230 bp give good targeting and integration efficiency [16]. Here, we created the *RAD52* deletion construct by the PCR amplification of a DNA fragment containing medium-sized homology regions of 226 bp upstream and 272 bp downstream *RAD52* flanking sequences (Figure 3A, Appendix A).

We used the method of targeted ends-out insertion mutagenesis as described for *N. castellii* to transform the strain YMC48 [16,22]. In this approach, the purified *RAD52* deletion construct is introduced into the cells by the lithium acetate-mediated transformation method and transformant colonies were initially screened by their ability to grow on minimal uracil dropout plates [39]. To confirm the replacement of the gene, we performed three different PCR reactions on genomic DNA extracted from the transformant colonies. First, a PCR targeting the upstream and downstream flanking regions of the *RAD52* locus was performed to identify the replacement of the *RAD52* gene (1356 bp) with the klURA3 cassette (1571 bp) (Figure 3A). To effectively target the *RAD52* locus, the primer set used binds to sequences further upstream and downstream from the homology regions used for the creation of the *RAD52* deletion construct. Because the klURA3 cassette is 215 bp larger than the RAD52 gene, we were able to evaluate the difference in amplicon size by resolving the PCR products with agarose gel electrophoresis and thus determined whether the sequence in the *RAD52* locus was replaced with the klURA3 cassette (2966 bp band) or not (2751 bp band) in the transformants (Figure 3B). The second and third PCR reactions were performed with primers that targeted sequences internal to the klURA3 cassette, in combination with the primers used to target the flanking sequences (Figure 3A). Because the internal primers are directed to the *K. lactis URA3* gene, this amplification reaction can only generate products for the *rad52Δ* strains. Coincidentally, the amplicon size for both the internal PCR reactions is the same (Figure 3C, 1952 bp band). 

From 203 colonies tested after transformation, we found 136 *rad52Δ* mutants with the correct target locus replacement (65.5% transformation efficiency). The large number of colonies obtained and the high transformation efficiency are comparable to what we have seen previously after transformation with long homology regions (>500 bp) [16,22]. This high efficiency with medium-sized homology regions could be due to the usage of *URA3* as a marker gene rather than antibiotic resistance. Our results indicate that efficient targeted insertion mutagenesis can be achieved in *N. castellii* strains utilizing the *URA3* marker gene and homology regions as short as 226 bp in length. 

In summary, we successfully deleted the *RAD52* gene in the *N. castellii* haploid background and replaced it with the klURA3 cassette. We were able to transform the cells with exceptionally high efficiency by using medium-sized homology regions for the homology-mediated gene replacement, further highlighting the utility of this molecular method in a non-conventional budding yeast. 

### 3.3. Characterization of N. castellii rad52Δ Mutants

*RAD52* is a non-essential gene in *S. cerevisiae* despite its central role in DNA damage repair [4]. We started the characterization of *N. castellii rad52Δ* mutant strains by evaluating their ability to grow in nutritive media and their colony and cell morphology (Figure 4). Colonies of *rad52Δ* cells grown on yeast rich media (YPD) for 2 days at 25 °C showed a wild-type morphology, with smooth, bright and light cream-colored colonies (Figure 4A) [19]. However, our colony measurements showed that *rad52Δ* colonies were, on average, slightly smaller than WT colonies (Appendix A). During the passaging of the cells on YPD plates for >300 generations, we did not observe any growth defect or any significant changes in the colony morphology (Appendix A). Additionally, we tested the viability of the cells by performing a colony forming unit count and found a small difference in viability between the *rad52Δ* strains and the WT (Appendix A). These data suggest that the loss of the *RAD52* gene function does not affect the viability of the *N. castellii* cells. 

Interestingly, *rad52Δ* cells showed a more spherical shape and were determined to have an average size of 4.07 × 3.33 μm, which was larger than that of their parental strain, 3.68 × 3.03 μm (Figure 4B). Further analysis showed that when the cell size measurements were divided into four 1 µm size categories, the distribution of the *rad52Δ* data points was shifted towards the larger size for both the long and short axis (Figure 4C). This “bloating” phenotype was, to our knowledge, not reported for *S. cerevisiae rad52* mutant strains until recently, when it was described that *rad52* mutant cells are on average larger than WT cells [40]. Changes in cell size have been correlated to changes in the dynamics of the cell cycle; thus, we could hypothesize that the increased cell size and decrease in the growth rate of *N. castellii rad52Δ* mutants is an indication of a slower transition through the cell cycle, possibly due to unrepaired DNA damage in the cell, as recently demonstrated in *S. cerevisiae rad52* knockout strains [40,41]. 

To further evaluate their growth ability, we performed a growth assay in liquid YPD broth. Cultures were started at a density of 3 × 10^5^ cells/mL and incubated at 25 °C (Figure 4D). The growth of the cells was monitored by measuring the cell density every two hours for a total of 8 h. Interestingly, *rad52Δ* cells had a slightly lower growth rate than their parental WT strain. Indeed, by tracing the best fit trendline, we could estimate that the *rad52Δ* mutant strains required 145 min to double, while the WT strain had a doubling time of 127 min (Appendix A). We confirmed this observation when performing a 5-day serial passaging experiment of the cell cultures (Appendix A). The cultures were re-inoculated into fresh medium to the same starting density every day, and the cell density was measured after 24 h of growth. In this long-term growth analysis, *rad52Δ* cultures consistently achieved a lower cell density than WT cultures in every 24 h cycle These results are consistent with the lower doubling time observed in *S. cerevisiae rad52* mutant strains [40,42].

The optimal growth temperature of *N. castellii* cells has been reported to be 25 °C; however, cells can still grow at 20 and 30 °C [19]. We tested the growth of *rad52Δ* mutant strains at these three permissive growth temperatures in a spot assay performed on YPD plates. After 2 days of incubation at 30 °C, *rad52Δ* mutant colonies showed less growth than the parental strain; however, the viability of the cells remained unchanged after 3 days of growth, suggesting that *N. castellii rad52Δ* strains had impaired growth at higher temperatures (Figure 4E, Appendix A). A lower growth rate at elevated temperatures has also been shown for *S. cerevisiae rad52Δ* mutants [43]. Interestingly, *N. castellii rad52Δ* cells struggle to grow at 20 °C, as seen in our cell viability assay (Appendix A). 

In summary, *N. castellii rad52Δ* mutants are viable and able to survive for multiple generations in solid and liquid media. While *rad52Δ* mutants are able to form WT-like colonies, cells were shown to be, on average, larger and more spherical than WT cells. Moreover, *rad52Δ* mutant cultures have longer doubling times, a phenotype that is more notable when growing the cells at other permissive temperatures. These phenotypic differences might be related to a stalling in the G2 phase, as recently described for *S. cerevisiae knockout* mutants of the *RAD52* epistasis group [40]. 

### 3.4. N. castellii rad52Δ Mutants Are Sensitive to Genotoxic Agents

The first description of *RAD52* and *RAD52* epistasis group mutants was based on the sensitivity of these strains to X-ray damage and they were later described to be sensitive to multiple DNA-damaging agents as they have deficiencies in DNA repair mechanisms. To investigate the sensitivity of *N. castellii rad52Δ* mutants against DNA-damaging agents, we performed spot assays and compared the growth of several mutants to their parental strain (WT) on YPD after exposure to ultraviolet (UV) radiation or in the presence of the DNA-damaging agents hydroxyurea and bleomycin (Figure 5).

First, we exposed the cells to UV irradiation, which is a physical agent that can indirectly induce DSBs. UV at low exposures primarily creates pyrimidine dimers in the DNA. When left unrepaired, these dimers can generate replication stress, leading to the generation of DNA double-strand breaks (DSBs). At low irradiation of 50 J/m^2^, the *rad52Δ* mutants showed no growth defect when compared to the WT strain (Figure 5). Only when cells were exposed to higher irradiation (100 J/m^2^), a negative effect on the growth of the mutants was visible on the plates (Figure 4). Our results are consistent with what has been observed in *S. cerevisiae rad52Δ* mutants, which are hypersensitive to ionizing radiation but were only mildly sensitive to UV irradiation in a spot assay [44,45]. 

Next, we tested how the cell growth was affected in the presence of two DNA-damaging chemical agents. Hydroxyurea will deplete the dNTP pool of the cells, stalling replication forks and activating the S-phase checkpoint, therefore preventing cells from successfully completing DNA replication. The stress generated at collapsed replication forks can lead to the formation of DSBs, and thus hydroxyurea can indirectly induce DNA damage [46]. *S. cerevisiae rad52Δ* strains were sensitive to 50 mM hydroxyurea treatments in spot assays [45,47,48]. Likewise, our results indicate that *N. castellii rad52Δ* mutants are sensitive to hydroxyurea, showing greatly reduced growth capacity on plates containing 50–100 mM hydroxyurea (Figure 5). 

Bleomycin is known as a radiomimetic agent because, as gamma irradiation, it can directly cause DSBs at high cellular doses. *S. cerevisiae rad52Δ* strains are highly sensitive to bleomycin treatments, showing poor growth at 2 µg/mL and almost no growth at 3–4 µg/mL in spot assays [45,49]. Interestingly, the growth of *N. castellii rad52Δ* mutants was negatively impacted by the presence of 2 µg/mL bleomycin and the effect intensified at 3 µg/mL bleomycin, indicating that these strains are unable to repair the lethal damage caused by this chemical agent (Figure 5). 

To analyze the sensitivity to genotoxic agents more extensively, we tested the cytotoxicity of each treatment by obtaining survival curves, which provide quantitative data. WT and *rad52*Δ cell cultures were grown to the logarithmic phase and the cell density measured in a Bürker chamber. In parallel, cells were spread on YPD plates either without (control) or with the treatment of different amounts of UV irradiation, hydroxyurea or bleomycin. The number of colony forming units (CFU) was counted after 3 days of incubation at 25 °C and normalized to the control plates (Figure 6, Appendix A). Hence, the toxicity to each dosage of the treatment was quantified based on the decrease in viability of the cells in the presence of the treatment. 

The *rad52Δ* mutant cells showed a marked decrease in viability already at a UV dosage of 25 J/m^2^. At this dosage, only 41% of the *rad52Δ* cells retained the ability to grow and form colonies, compared to a 96% survival rate for the WT parental strain, thus showing that *rad52Δ* mutants have enhanced sensitivity to this physical agent (Figure 6A). At a dosage of 75 J/m^2^, the difference is less pronounced, with the *rad52Δ* mutants and WT showing survival rates of 24% and 4%, respectively. We conclude that *N. castellii rad52Δ* mutant cells exhibit increased sensitivity to UV irradiation compared to when they are exposed to this physical agent in the spot assay (Figure 5).

For the hydroxyurea treatments, a dosage of 45 mM hydroxyurea gave a significant decrease in viability for the *rad52Δ* mutants (Figure 6B). At this concentration, the *rad52Δ* mutants showed a survival rate of 46%, while the WT strain was unaffected by the presence of the compound. Interestingly, we observed that with the increase in dosage, the mutant colonies became smaller, an effect that was also visible for the WT strain at the highest dosage tested, although not as pronounced. 

When obtaining survival curves with radiomimetic compound bleomycin, we found that *rad52Δ* mutants were sensitive to low concentrations of bleomycin. The viability of the cells was reduced to 44% when exposed to 1 µg/mL, compared to the 90% survival rate of the WT strain, thus providing more evidence that *N. castellii rad52Δ* mutant strains are susceptible to DNA-damaging agents (Figure 6C). 

Notably, for all treatments, we observed that the cells were more susceptible when spread as single colonies than in the spot assay, as they showed a pronounced negative effect on viability at lower dosages than previously tested. This difference between treatment conditions in the methodologies used could be an effect of the penetrance of the treatments from a community of cells in the spot assay to the single cell in the viability assay. 

In conclusion, here, we identified the *RAD52* gene in the budding yeast *N. castellii* and investigated the importance of Rad52 for the repair of DNA damage in this species. Notably, we determined that the Rad52 protein is highly conserved in *N. castellii* and the comparison of the primary amino acid sequence against *S. cerevisiae* Rad52 showed that all essential residues needed for the protein function are conserved. Through targeted gene replacement, we were able to create *rad52Δ* null mutants that were shown to have lower growth capacities and different morphologies from their parental wild-type strains. Importantly, the decreased viability of *N. castellii rad52Δ* mutants in the presence of genotoxic and DNA-damaging agents reflects that the primary role of the Rad52 protein in the repair of DNA damage has been conserved. 

The identification of homologs in non-conventional species is widening our understanding of the evolution of gene function. The Rad52 protein has been shown to be part of several different cellular processes [50]. Moreover, *RAD52* is required for telomere maintenance in the absence of telomerase in *S. cerevisiae*, by promoting break-induced replication between telomeres. In the absence of telomerase, *N. castellii* cells effectively activate an alternative lengthening of telomeres (ALT) mechanism to maintain short telomeres that are preceded by a short subtelomeric element [51]. In the future, we would like to evaluate whether this mechanism also requires *RAD52* to guarantee the survival of the cells in the absence of telomerase.

## Figures and Tables

**Figure 1 genes-14-01908-f001:**
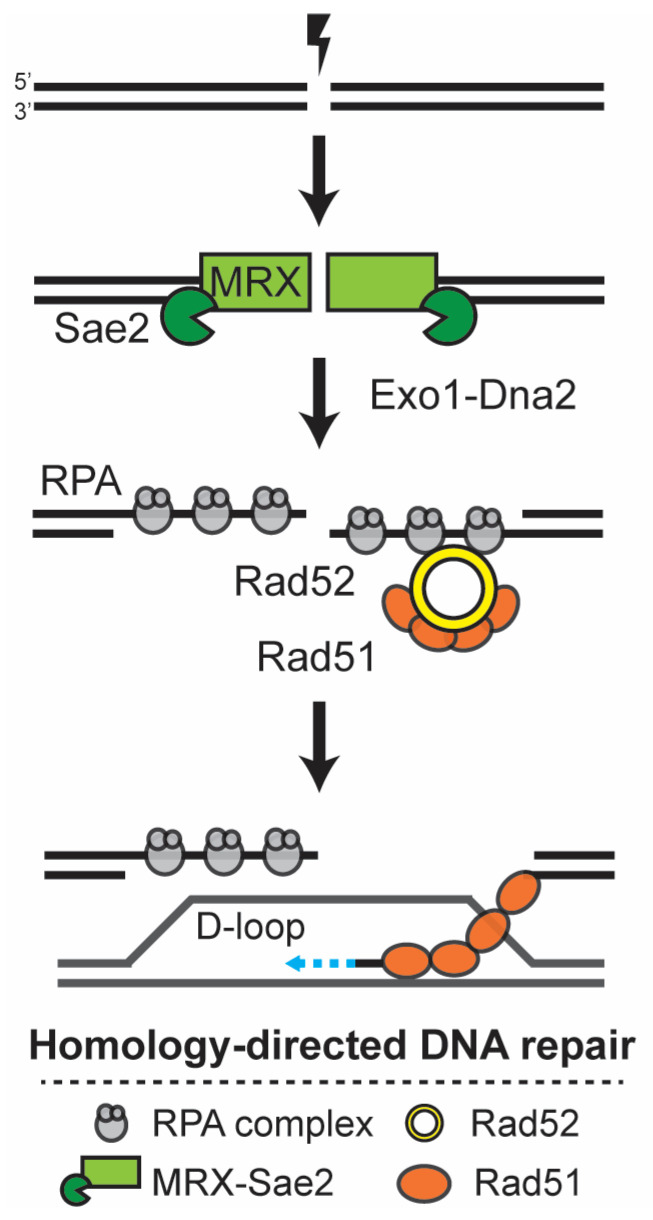
Simplified overview of the Rad52-dependent DNA double-strand break repair mechanism. DNA double-strand breaks (DSBs) are toxic lesions to the cell, which are preferentially repaired by homology-directed DNA repair. During the S and G2 phases, each side of the break is initially processed by Mre11-Rad50-Xrs2 (MRX) together with the Sae2 nuclease, and later by Exo1-Dna2 nucleases, generating ssDNA that is quickly bound by Replication Protein A (RPA). Rad52 facilitates the formation of the Rad51 filament at the ssDNA, and this filament promotes the homology search and invasion of the donor template, forming a D-loop. After the copying of the donor template by the DNA polymerase, the ends are ligated and the structure resolved.

**Figure 2 genes-14-01908-f002:**
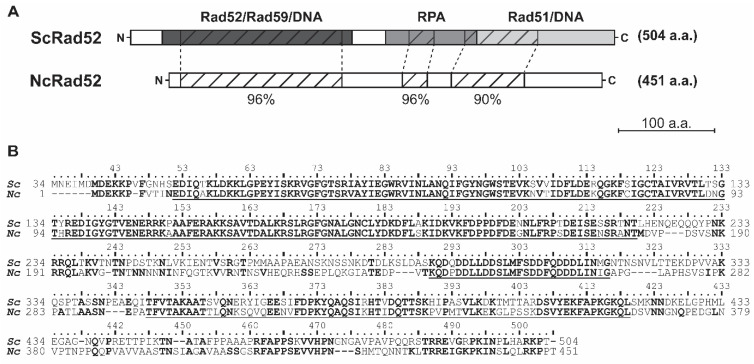
Amino acid sequence comparison of *S. cerevisiae* and *N. castellii* Rad52 proteins. The sequence of *N. castellii RAD52* was retrieved from the KEGG genome database and aligned to the sequence of *S. cerevisiae* S288C Rad52. The N-terminal amino acids 1–33 of *S. cerevisiae* Rad52 are not required for in vivo function and thus were omitted from the analysis [26]. (**A**) Schematic representation of *S. cerevisiae* and *N. castellii* Rad52 protein comparison. The two proteins share overall 70.3% sequence similarity. Three highly conserved regions were identified and mapped to the functional domains of *S. cerevisiae* Rad52: 96% similarity at the N-terminal domain (amino acids 52-220 of *S. cerevisiae* Rad52), 96% similarity at the RPA-binding domain (amino acids 290-316 of *S. cerevisiae* Rad52) and 90% similarity at the C-terminal domain (amino acids 348-419 of *S. cerevisiae* Rad52). (**B**) The protein sequences of *S. cerevisiae* Rad52 (*Sc*) and *N. castellii* Rad52 (*Nc*) were aligned using the ClustalW algorithm. Identical amino acid residues are in bold for both proteins, while similar amino acids are only in bold in the *N. castellii* sequence. The highly conserved regions depicted in (**A**) are underlined.

**Figure 3 genes-14-01908-f003:**
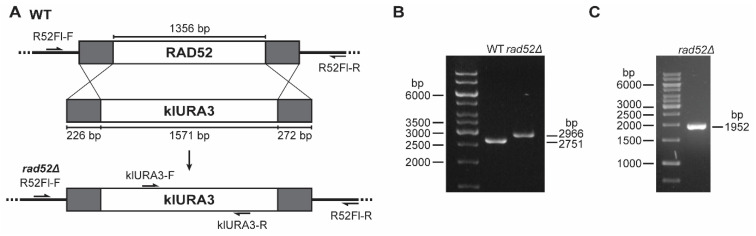
Deletion of the *N. castellii RAD52* gene by gene replacement. (**A**) Schematic representation of the replacement of the *RAD52* gene with the klURA3 cassette. The genomic structure of the *N. castellii RAD52* locus in wild-type strains (WT) contains the 1356 bp ORF encoding the gene of interest. The klURA3 cassette (1571 bp) flanked by upstream and downstream sequences of homology (grey) was transformed into the cells by the lithium acetate-mediated transformation method. Recombination between the homology regions allows for the replacement of the *RAD52* gene with the klURA3 cassette, effectively deleting the *RAD52* gene (*rad52Δ*). Primers used for the screening are indicated. (**B**) Genotype analysis of transformant colonies by flanking PCR. Agarose gel electrophoresis of the amplicons generated after PCR with primers targeting the *RAD52* gene flanking regions (R52Fl-F and R52Fl-R). The *rad52Δ* locus gives a slightly longer PCR product (2966 bp) than the WT (2751 bp). (**C**) Genotype confirmation of *rad52Δ* mutants by internal PCR. Agarose gel electrophoresis of the amplicons generated after PCR performed with primers that bind the klURA3 cassette (klURA3-F) in combination with the flanking primer R52Fl-R. The 1 kb GeneRuler (Thermo Scientific) was used as a DNA marker in all agarose gels.

**Figure 4 genes-14-01908-f004:**
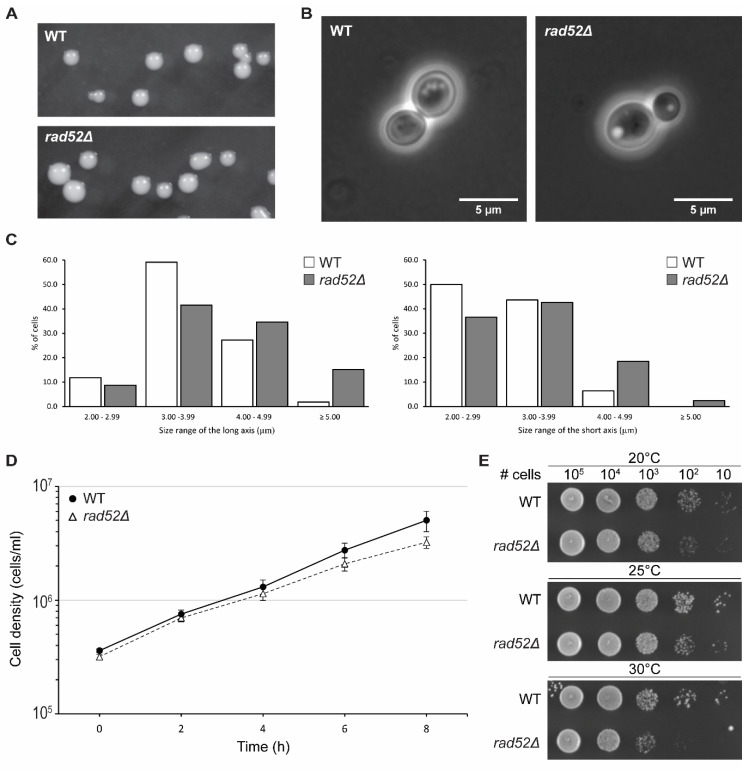
Characterization of *N. castellii rad52Δ* mutants. Comparison of WT (YMC48) and *rad52Δ* mutant strains. (**A**) WT and YMC490 (*rad52Δ*) colonies were photographed after 2 days of growth on YPD plates. (**B**) Phase contrast microscopy of WT and *rad52Δ* mutants. Cells from an overnight culture were imaged with a Zeiss Observer Z1 inverted microscope Ph3 stage. (**C**) Cell size measurement comparison of WT and *rad52Δ* mutants. Measurements were taken across the long and short axis of the cell’s elliptical shape and the values were separated into four size categories. For each category, the frequency of the length values is represented as a percentage of the total values for the respective axis for WT (white box, n = 110) and *rad52Δ* (grey box, n = 298) cells. (**D**) Growth analysis of *rad52Δ* mutants in liquid YPD media. Cultures of WT and *rad52Δ* strains were inoculated to a cell density of 3 × 10^5^ cells/mL and grown for 8 h at 25 °C. The cell density was measured every two hours using a Nucleocounter YC-100. The graph represents the average cell density values of three biological replicates; error bars indicate SEM for all replicates (n = 3). (**E**) Temperature sensitivity spot assay for the *rad52Δ* strain YMC490. Tenfold serial dilutions of WT and *rad52Δ* strains were spotted on YPD media at the indicated cell amounts. Cells were grown for 2 days at 20, 25 and 30 °C as indicated.

**Figure 5 genes-14-01908-f005:**
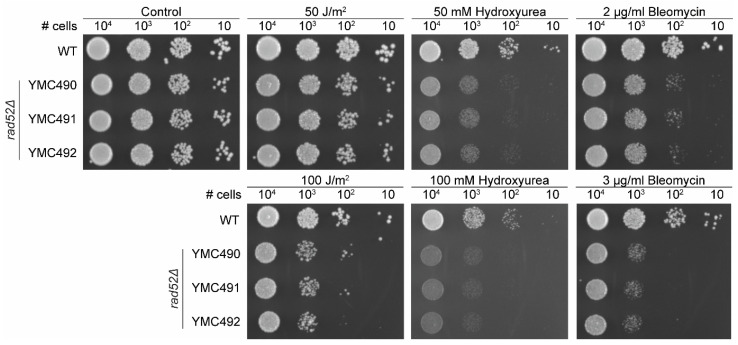
*N. castellii rad52Δ* mutants are sensitive to DNA-damaging agents. Spot assays under different conditions were performed for the *rad52Δ* mutants. Tenfold serial dilutions of WT (YMC48) and three different *rad52Δ* strains (YMC490, YMC491 and YMC492) were spotted on YPD plates, in the absence or presence of either 50–100 J/m^2^ UV irradiation, 50–100 mM hydroxyurea or 2–3 μg/mL bleomycin as indicated. Non-treated control plate indicated as Control. Cells were incubated for 3 days at 25 °C.

**Figure 6 genes-14-01908-f006:**
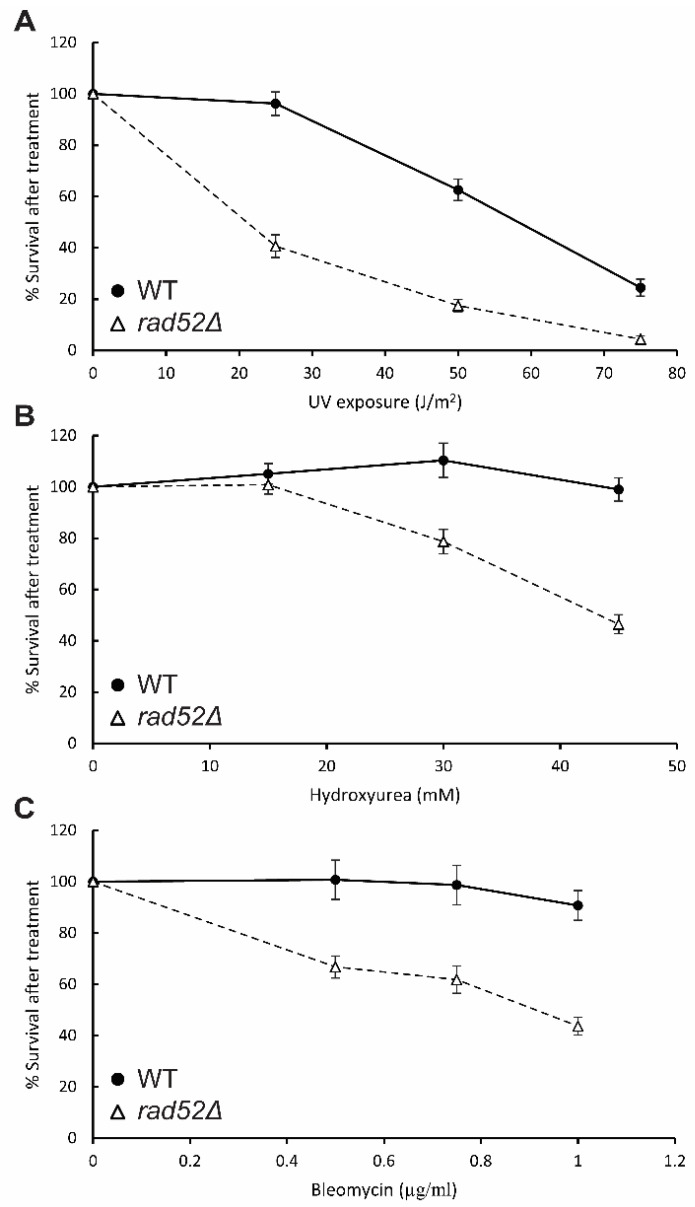
Quantitative survival curves of *N. castellii rad52Δ* strains treated with genotoxic agents. Cells from WT and 3 different *rad52Δ* strains (YMC490–492) were spread onto YPD solid media and exposed to different doses of UV irradiation (**A**), or YPD solid media containing hydroxyurea (**B**) or bleomycin (**C**) at different concentrations. After incubation for 3 days at 25 °C, the number of colony forming units was counted for each treatment and normalized to the non-treated control plate. Bars represent SEM for n = 3 (WT) and n = 9 (*rad52Δ*).

**Table 1 genes-14-01908-t001:** *Naumovozyma castellii* strains used and created in this study.

Strain	Genotype	Parental Strain	Reference
YMC48	*MATα*, *hoΔ::hphMX4*, *ura3*	YMC25	Karademir Andersson et al., 2016 [16]
YMC490-494	*MATα*, *hoΔ::hphMX4*, *ura3*, *rad52Δ::klURA3*	YMC48	This study

## Data Availability

Materials and strains are available upon request.

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
