# Peer review of "Characterization of the RAD52 Gene in the Budding Yeast Naumovozyma castellii"

_genes, 2023, doi:10.3390/genes14101908_

Round 1
Reviewer 1 Report
Review of manuscript: “ Characterization of the RAD52 gene in the budding yeast Naumovozyma castellii” by Itriago et al.
This manuscript presents the characterization of the RAD52 gene from N. castelli and its KO mutant. The manuscript is clearly written, the results are very straightforward and the data presented is solid.
I have only some suggestions for data that could make this paper more informative to people working on DNA repair in other organisms:
1) Rad52 is regulated by SUMOylation in S. cerevisiae. Are the SUMOylation sites conserved in N. castelli?
2) In S. cerevisiae, strains deleted for RAD52 demonstrate high frequency of petit generation (probably due to loss of mitochondrial DNA due to lack of DNA repair). Is that also trie in N. castellii?
Minor corrections:
Line 59: SSA is considered “error prone” only because its end-result is a deletion. But being homology-based, it is as error-free as SDSA or DSBR. So perhaps “error prone” is not a good qualification in the Intro context.
Line 258: “was”, not “is”
Lines 312, 314, 340, 346: The word “medium” appears here a lot, and its meaning is not clear until the end. I assume that in the lab the authors use “long”, “medium” and “short” stretches of homology, but the meaning of “medium” in the sentences mentioned remains obscure. Perhaps the authors should rephrase.
Reviewer 2 Report
The manuscript describes the RAD52 gene of the budding yeast Naumovozyma castellii. The gene was defined, deletion mutant created, and initially characterized (cell morphology, growths, sensitivity to three DNA damaging agents). In addition, based on a well-done bioinformatics analysis of similarity to S. cerevisiaeprotein, the authors infer that the protein might play a mediator role in recombination. The experimental proof of the postulated role will be the subject of further experiments. N. castellii is a model in telomere biology, but, unfortunately, the role of Rad52 in telomere maintenance is not investigated here. The manuscript could be regarded as a short communication with a moderate descriptive advance. The significance of studies of Rad52 in the particular yeast needs to be better articulated by the authors.
Minor comments.
The introduction is somewhat out of focus. Detailed recombination mechanism descriptions are optional for specialists and challenging to understand for non-specialists. The solution will be a simple Figure with the main players in cartoon form. Also, the elaborate description was not relevant to the results of the work because the mentioned mechanisms were not investigated directly. Instead, it would also be good to tell more about N. castellii and better justify the study, telling more about telomere biology and the role of Rad52 in it. What is known about recombination in this yeast?
Lines 31, Definition of error-prone and error-free are imprecise; it is well-known that HR is associated with elevated mutagenesis PMID: 7672595, 23146099.
Lines 64 and 223-224 convey partially conflicting information.
Line 74. Elaborate on “processive telomerase activity”.
Line 84. HU causes DSB indirectly (as the authors correctly explain later), but here, the statement sounds like HU induces them directly.
Paragraph between Lines 239-252. The word combinations "nothing is surprising" or "not strange" do not support readers' desire to learn something novel from the study.
Line 333. Still, wt control would be reasonable to include in Fig. 2C.
Lines 348-7. What is the benefit of studies in non-conventional yeast? Was the method of homology-based disruption successfully used for the first time, justifying a lengthy description of simple experiments?
Lines 383-385. It would be good to see the author's opinion of the mechanism of the changes.
Lines 388-390, Lines 418-420. Should a cell cycle analysis be done to explore that?
Lines 443-445. Later quantitative experiments do not collaborate with this statement (Lines 483-484).
Line 472 (Fig. 5). The sensitivity of the deletion strains to all three agents is similar, contrary to the results of qualitative tests (Fig. 4). The authors do not discuss the discrepancy.
Line 501. The authors did not directly investigate the mediator properties of N. castellii Rad52.
SupplementalFigs 5 and 8. Are the differences statistically significant?
